Resource

# Improving the computation efficiency of polygenic risk score modeling: faster in Julia

Annika Faucon[1,*], Julian Samaroo[2,*], Tian Ge[3], Lea K Davis[1,4], Nancy J Cox[1,4], Ran Tao[1,5], Megan M Shuey[1,4]

**To enable large-scale application of polygenic risk scores (PRSs) in a computationally efficient manner, we translate a widely used PRS construction method, PRS–continuous shrinkage, to the Julia programming language, PRS.jl. On nine different traits with varying genetic architectures, we demonstrate that PRS.jl maintains accuracy of prediction while decreasing the average runtime by 5.5×. Additional programmatic modifications improve usability and robustness. This freely available software substantially improves work flow and democratizes usage of PRSs by lowering the computational burden of the PRS–continuous shrinkage method.**

## Introduction

The conceptual framework known as the "liability-threshold model" asserts that complex diseases have many contributing variants of small effect, which collectively contribute to a continuous distribution of genetic liability in a population. Thus, when a large-enough collection of risk alleles is aggregated in an individual together with environmental risk factors such that they pass a critical threshold, the complex disease will manifest (Falconer, 1965). The additive genetic portion of this liability attributable to common variants can be estimated with a polygenic risk score (PRS). A PRS is generally calculated as a weighted sum of risk alleles present in an individual genome, where the weights are defined by the effects estimated in genome-wide association studies (GWASs) (Chatterjee et al, 2016).

Since the advent of PRS methods, various studies have proven their potential to improve health by informing therapeutic intervention (Tikkanen et al, 2013; Mega et al, 2015), disease screening (Hsu et al, 2015), and lifestyle choices for a multitude of polygenic conditions. In fact, polygenic risk scoring has long been at the center of genetic research (MultiBLUP [Speed & Balding, 2014], PLINK [Purcell et al, 2007], PRSice [Choi & O'Reilly, 2019], LDpred

[Vilhjalmsson et al, 2015]). In simulation and real data analyses, PRS–continuous shrinkage (CS) was demonstrated as a top-performing method (Ge et al, 2019; Pain et al, 2021). Despite their popularity and importance, PRS methods need development, particularly related to computational expense. As large datasets become publicly available and computation moves to the cloud (Langmead & Nellore, 2018), research demands the use of computational programs that can scale and cost-effectively use resources. Because the Julia programming language has consistently demonstrated increased efficiency of computation over other programming languages, along with other advantages (Bezanson et al, 2018), we created a Julia translation of the commonly used Python based PRS-CS program, PRS.jl.

Below, we introduce the PRS.jl program and benchmark it against PRS-CS, tracking model accuracy and computational improvements across nine well-characterized polygenic phenotypes including both continuous and binary outcomes.

## Results

### PRS.jl performance overview

PRS.jl is a direct translation of the PRS-CS Python program into the Julia language. This translation improves the computational efficiency of PRS estimation across a variety of polygenic traits.

Using the auto global shrinkage calculation with 10,000 MCMC iterations on a single Haswell node, eight CPUs available total, and a maximum memory allocation of 80 GB, we observed an average 5.5× improvement in computational speeds when using PRS.jl compared with PRS-CS across nine phenotypes. The improvements in speed ranged from 3.8× to 6.4× (Table 1). For the quantitative phenotypes – body mass index, high-density lipid cholesterol, low-density lipid cholesterol, total cholesterol, triglycerides, and estimated glomerular filtration rate – the average improvement was 5.6×. For the binary traits – asthma, coronary artery disease, and type 2 diabetes mellitus – the improvement was 5.1×. These

[1]Vanderbilt Genetics Institute, Vanderbilt University Medical Center, Nashville, TN, USA   [2]JuliaLab, Massachusetts Institute of Technology, Boston, MA, USA   [3]Psychiatric and Neurodevelopmental Genetics Unit, Center for Genomic Medicine, Massachusetts General Hospital, Boston, MA, USA   [4]Division of Genetic Medicine, Vanderbilt University Medical Center, Nashville, TN, USA   [5]Department of Biostatistics, Vanderbilt University Medical Center, Nashville, TN, USA

Correspondence: megan.m.shuey.1@vumc.org
*Annika Faucon and Julian Samaroo contributed equally to this work.

**Table 1.   Individual and average runtimes for polygenic risk score–continuous shrinkage (PRS-CS) and PRS.jl by phenotype.**

| | PRS-CS | | | | PRS.jl | | | | Average |
|---|---|---|---|---|---|---|---|---|---|
| | Run 1 | Run 2 | Run 3 | Mean (SD) | Run 1 | Run 2 | Run 3 | Mean (SD) | Improvement[a] |
| Quantitative phenotypes | | | | | | | | | |
| Body mass index | 62:21:25 | 65:12:16 | 59:41:13 | 62:24:58 (1:28:32) | 12:01:15 | 12:20:31 | 11:19:32 | 11:53:46 (0:31:10) | 5.3 |
| Cholesterol | 56:51:43 | 55:36:35 | 59:12:59 | 57:13:46 (1:49:52) | 12:07:50 | 8:15:23 | 12:34:38 | 10:59:17 (2:22:34) | 5.2 |
| eGFR | 65:52:20 | 69:43:40 | 77:57:50 | 71:11:17 (6:10:36) | 13:54:28 | 15:57:40 | 14:01:35 | 14:37:54 (1:09:10) | 4.9 |
| High-density lipoprotein | 59:58:13 | 56:39:56 | 59:19:08 | 58:39:06 (1:45:02) | 8:21:45 | 8:16:13 | 10:58:48 | 9:12:15 (1:32:19) | 6.4 |
| Low-density lipoprotein | 63:38:06 | 56:17:24 | 56:47:47 | 58:54:26 (4:06:08) | 8:26:04 | 8:18:47 | 12:54:02 | 9:52:58 (2:36:51) | 6.0 |
| Triglycerides | 58:00:52 | 56:04:48 | 60:45:49 | 58:17:10 (2:21:13) | 8:18:01 | 8:09:24 | 11:48:25 | 9:25:17 (2:04:02) | 6.2 |
| Binary phenotypes | | | | | | | | | |
| Asthma | 41:58:26 | 40:15:14 | 41:29:02 | 41:14:14 (0:53:10) | 5:26:44 | 6:25:06 | 7:27:54 | 6:26:35 (1:00:36) | 6.4 |
| Coronary artery disease | 66:59:45 | 66:00:16 | 64:05:35 | 65:41:52 (1:28:32) | 10:43:53 | 15:05:15 | 13:04:09 | 12:57:46 (2:10:48) | 5.1 |
| Type 2 diabetes mellitus | 68:35:38 | 69:10:26 | 68:08:43 | 68:38:16 (0:30:57) | 17:03:18 | 20:13:24 | 16:38:33 | 17:58:25 (1:57:33) | 3.8 |
| All phenotypes combined | | | | | | | | | |
| | 544:16:28 | 535:00:35 | 547:28:06 | 542:15:03 (6:28:16) | 96:23:18 | 103:01:43 | 110:47:36 | 103:24:12 (7:12:35) | 5.5 |

All runtimes are presented as hour:minutes:seconds.
[a]Average improvement is estimated as the mean PRS-CS/mean PRS.jl.

reported computational times represent the total amount of runtime when chromosomes are sequentially analyzed; processing the chromosomes in parallel can substantially reduce time to results.

We next demonstrate that these improvements in speed did not come at the expense of PRS accuracy (Fig 1).

Next, we show the retained accuracy of PRS.jl estimate for given phenotypes by demonstrating the consistency of posterior SNP weights and the resulting PRSs compared with PRS-CS. Specifically, to assess the consistency of SNP weights, we calculate the squared error for each SNP between the PRS.jl and PRS-CS output (Table 2). The median squared error between the two algorithms ranged from $2.00 \times 10^{-11}$ to $6.83 \times 10^{-11}$ across phenotypes, with a median of $3.00 \times 10^{-11}$. This is similar to the median squared errors within the same program on different runs (Table S1). A $t$ test comparing the posterior SNP effect sizes estimated by PRS.jl and PRS-CS found no statistically significant difference (all $P > 0.85$).

We examined accuracy of the PRSs relative to the traits measured in BioVU. For the quantitative traits, we compared prediction accuracy by $R^2$ between the observed and predicted phenotypes in the BioVU testing set (Table 3). For the binary traits, we compared prediction accuracy by area under the curve (AUC), the Nagelkerke $R^2$, and the odds ratio of the top 10% versus the remaining 90% (Table 4). PRS-CS and PRS.jl had nearly identical accuracies across all tested traits.

Lastly, because performance can vary based on sample size, we also provide an estimate of performance for the most common binary, asthma, and continuous phenotype, triglycerides, for three different sample sizes (Table S2). As expected, these runs took longer and had lower $R^2$ than the previous runs that used the largest number of samples. Regardless of sample size, however, the PRS-CS and PRS.jl programs had nearly identical accuracies for

the traits with similar sample sizes and consistent runtime improvements with PRS.jl.

## Discussion

PRSs are hailed for their potential to revolutionize clinical and precision medicine (Torkamani et al, 2018; Reay et al, 2020). Despite early successes there remain considerable concerns relating to the broader applicability of PRSs to genetically diverse populations and the computational power required to use these approaches at scale. With the growing availability of large-scale biobanks including All of Us (All of Us Research Program Investigators et al, 2019), Biobank Japan (Nagai et al, 2017), FinnGen (Kurki et al, 2022 *Preprint*), and UK Biobank (Bycroft et al, 2018), the need for improved genomic analysis tools that have the potential to handle these larger sample sets in a faster, less computationally intensive manner without sacrificing efficacy is paramount. The Julia programming language has many features that allow for these improvements including an efficient type-system and multiple dispatch, a variety of optimized matrix routines, and a straightforward application programming interface for accessing single instruction, multiple data (SIMD), and multi-threading. Specifically, the efficient type-system together with multiple dispatches means that the right version of functions can be called on in a computational manner that does not require checking types at runtime. Optimized matrix routines can also prevent excess memory usage, for instance, in the linkage disequilibrium calculation where a symmetric matrix type can be used. We utilize Julia's multi-threading capabilities to provide speedups beyonds those available from using multithreading in basic linear algebra subprograms (BLAS), particularly in the MCMC implementation. Indeed, computation efficiency profiling, where we

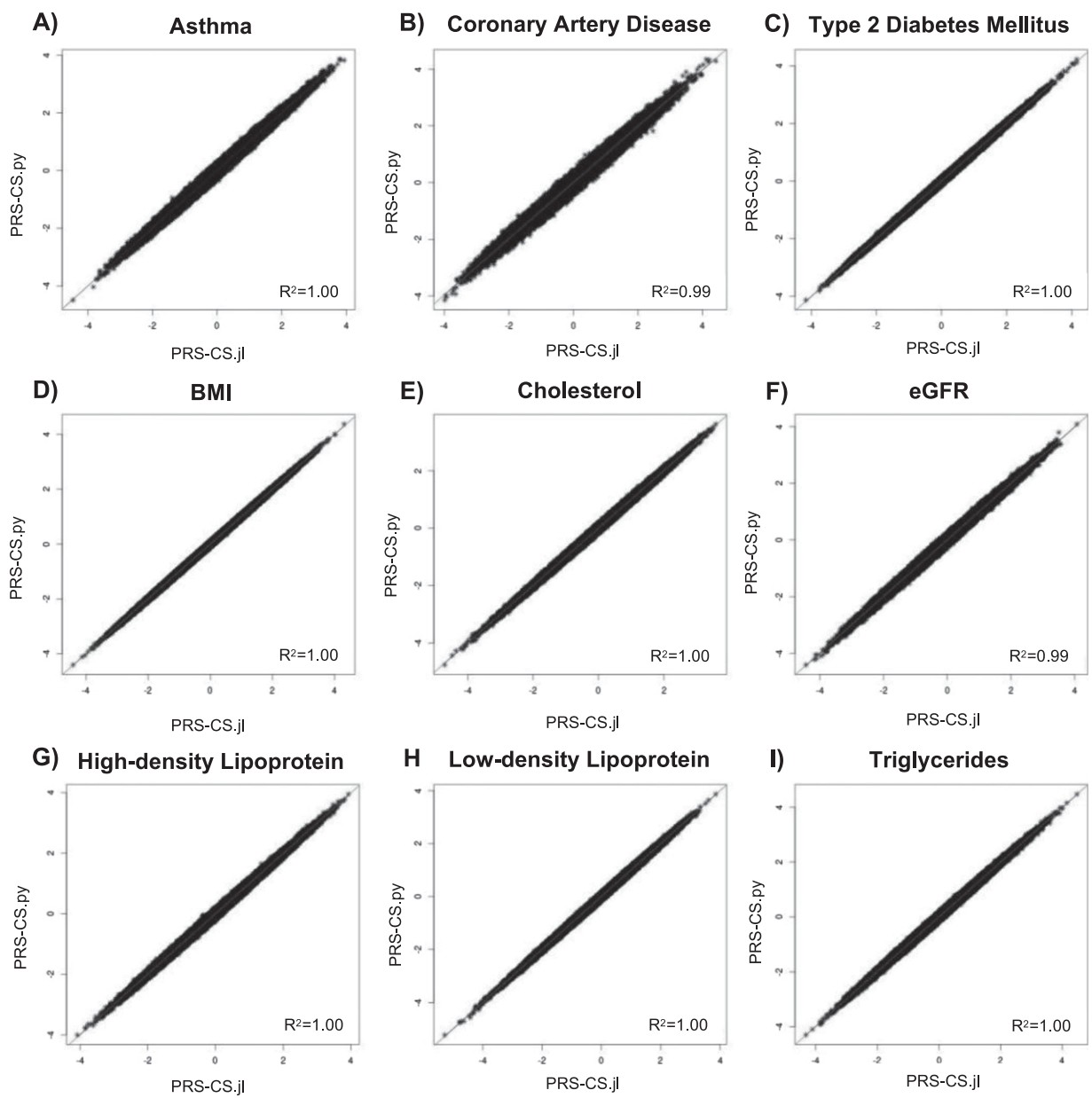

**Figure 1. Plots comparing polygenic risk score–continuous shrinkage (PRS-CS) and PRS.jl PRS estimates for each trait.**
**(A, B, C, D, E, F, G, H, I)** Plots of the PRSs calculated by the python implementation of PRS-CS (PRS-CS.py) on the y-axis compared with the scores calculated by PRS.jl on the x-axis for each trait: (A) asthma, (B) coronary artery disease, (C) type 2 diabetes mellitus, (D) body mass index, (E) cholesterol, (F) estimated glomerular filtration rate, (G) high-density lipoprotein, (H) low-density lipoprotein, and (I) triglycerides. The correlation $R^2$ are presented in the corner of each plot.

estimate the memory and CPU usage in individual runs for both the original (Python) implementation and the proposed (Julia) variant, demonstrates that these multi-threading capabilities have the capacity to drive the large speedups.

On the asthma benchmark, Julia uses a maximum of ~8 GB of memory, whereas Python used a maximum of ~6 GB of memory. For triglycerides, Julia and Python use approximate maximums of 5 and 2.5 GB, respectively. This difference may also be due in part to how memory is managed in each language. Julia uses a garbage collector (GC) to manage and collect user-allocated data, whereas Python uses a mix of reference counting (Refcounting) and garbage

collection. Refcounting is a deterministic mechanism, allowing data to be freed almost exactly as soon as it can be proven to be no longer used; GC, on the other hand, more lazily (and generally stochastically) frees data, because of tradeoffs inherent to GC design. However, GC trades higher overall memory usage for generally better performance of user code. As such, the Julia code which allocates the same amount of memory as the Python code has the potential to execute more efficiently. Thus, even though Julia uses more memory for the same numerical computations, it is expected that Julia's performance advantage over python is partially because of the usage of a GC instead of Refcounting. In

**Table 2. Median squared error and *P*-value from the *t* test comparing SNP weights between python and Julia implementations for a single run.**

| | | | SNP count | |
|---|---|---|---|---|
| | Median square error | *P*-value | GWAS | Polygenic risk score |
| Asthma | $2.07 \times 10^{-11}$ | 0.89 | 8,270,130 | 494,889 |
| Body mass index | $6.55 \times 10^{-11}$ | 1.00 | 2,529,253 | 719,311 |
| Coronary artery disease | $6.83 \times 10^{-11}$ | 0.91 | 8,440,435 | 782,510 |
| eGFR | $3.22 \times 10^{-11}$ | 0.93 | 17,393,472 | 774,105 |
| HDL | $3.29 \times 10^{-11}$ | 0.95 | 2,433,797 | 696,196 |
| LDL | $3.19 \times 10^{-11}$ | 0.97 | 2,424,334 | 695,115 |
| Total cholesterol | $3.25 \times 10^{-11}$ | 0.87 | 2,433,332 | 696,147 |
| Triglycerides | $3.17 \times 10^{-11}$ | 0.86 | 2,425,960 | 695,255 |
| Type 2 diabetes mellitus | $2.00 \times 10^{-11}$ | 0.85 | 35,369,247 | 780,627 |

eGFR, estimated glomerular filtration rate; GWAS, genome-wide association study; HDL, high-density lipoprotein; LDL, low-density lipoprotein; SNP, single nucleotide polymorphism.

**Table 3. Comparison of polygenic risk score–continuous shrinkage (PRS-CS) and PRS.jl performance for quantitative traits using, as covariates, age, sex, and PCs 1–10.**

| | $R^2$ | | |
|---|---|---|---|
| | PRS-CS | PRS.jl | Number of subjects |
| Body mass index | 0.1141(<0.0001) | 0.1141 (<0.0001) | 60,584 |
| Cholesterol | 0.1089 (0.0002) | 0.1088 (0.0002) | 34,347 |
| eGFR | 0.4921 (<0.0001) | 0.4922 (<0.0001) | 34,797 |
| High-density lipoprotein | 0.2326 (<0.0001) | 0.2326 (<0.0001) | 33,338 |
| Low-density lipoprotein | 0.0835 (<0.0001) | 0.0835 (0.0001) | 32,061 |
| Triglycerides | 0.0657 (<0.0001) | 0.0657 (<0.0001) | 34,531 |

All data is presented as mean (SD).

**Table 4. Comparison of polygenic risk score–continuous shrinkage (PRS-CS) and PRS.jl performance for binary traits.**

| | Nagelkerke $R^2$ | | Area under the curve | | 10% odds ratio | | Number of subjects | |
|---|---|---|---|---|---|---|---|---|
| | PRS-CS | PRS.jl | PRS-CS | PRS.jl | PRS-CS | PRS.jl | Cases | Controls |
| Asthma | 0.0176 (<0.0001) | 0.0176 (<0.0001) | 0.560 (<0.0001) | 0.560 (0.0002) | 1.54 (0.02) | 1.54 (0.02) | 8,210 | 64,618 |
| Coronary artery disease | 0.3212 (0.0001) | 0.3214 (0.0001) | 0.552 (0.0004) | 0.552 (0.0005) | 1.71 (0.02) | 1.73 (0.01) | 16,807 | 56,021 |
| Type 2 diabetes mellitus | 0.1715 (<0.0001) | 0.1716 (<0.0001) | 0.626 (0.0001) | 0.626 (0.0001) | 2.75 (0.017) | 2.77 (0.008) | 13,688 | 59,140 |

All data is presented as mean (SD).

additional, Julia's GC uses heuristics based on total available system memory to determine when to pause execution and initiate expensive GC scans; therefore, maximum memory usage measurements are not reliable predictors of memory requirements and will vary based on the amount of memory available on the system being used.

On the benchmark for average CPU usage, of the theoretical 800% usage possible with eight cores, Julia averages ~700%, whereas Python averages only about 300%. Both PRS.jl and PRS-CS use the same multi-threaded BLAS (openBLAS) to efficiently execute BLAS operations. Without additional speedups, PRS.jl would likely achieve only similar performance. However, PRS.jl uses additional multi-threaded mechanisms to achieve further speedups. Part of those speedups come from CSV reading (implemented in the CSV.jl package), although those improvements are limited to the relatively short CSV parsing phase. Most of the speedup is expected to come from multi-threading of the MCMC algorithm, which has many parallelizable regions of calculations. Specifically, updates of many model parameters are all executed with multi-threading, minimizing the span of non-scaling single-threaded regions of the execution.

Here, we demonstrate how a basic port of the commonly used PRS-CS package to the Julia language, PRS.jl, can improve the program's speed without sacrificing PRS accuracy across a variety of

traits. PRS.jl is freely available for download through GitHub (github.com/fauconab/PolygenicRiskScores.jl) and is a drop-in replacement for PRS-CS. Small usability improvements were made that allow the user to supply summary statistics with columns in any order and allows the user to specify supplied column names. No major changes to the algorithm code were made. Thus, the improvements reflect advantages of the Julia programming language over Python.

The available README text instructs even novice Julia programming language users how to execute this software with ease. Because of the usability and performance improvements, we believe PRS.jl will allow for broader and more efficient use of PRSs in genomic medicine.

Furthermore, the development of tools for genomic analyses that are both fast and computationally efficient, such as PRS.jl, have the potential to democratize genomic research. To date, human genomics research is overrepresented by high-income countries, which tend to have more powerful computational resources and greater funding for the sciences. The lack of population diversity and global representation is driven by many factors; however, a lack of resources, financial and human, are consistently noted as key limitations that prevent middle- and low-income countries from fully using and contributing to genomic research (Marques-de-Faria et al, 2004; Hardy et al, 2008; Seguin et al, 2008; Kaur et al, 2019). Because of this disparity, these countries could benefit the most from advances in genomic medicine.

Knowing the impact, utility, and potential of PRSs to drive personalized medicine while acknowledging the immense bias in data availability and usage, the National Institutes of Health funded a large initiative to fund PRS research in diverse populations. By reducing the computational needs of key algorithms, low resourced research groups can use these algorithms to benefit their scientific endeavors and provide potential benefit for their populations. The current versions of PRS-CS and PRS.jl are limited in their utility in ancestrally diverse populations (Duncan et al, 2019), a limitation that has been addressed by PRS-CSx (Ruan et al, 2022). Future work to extend Julia improvements to the PRS-CSx framework for faster trans-ancestry PRS calculations has been planned.

The PRSs in this set of work are derived using standard inputs and publicly available summary statistics. Different discovery GWASs for the same traits can provide different polygenic risk estimates at the individual level (Schultz et al, 2022); therefore, association of scores to clinical values in this paper may be different than other papers using similar clinical traits because of differences in the discovery summary statistics. As such, the PRSs generated in this paper do not represent the most optimized PRS for any particular trait. Despite this limitation in study design, our work clearly demonstrates that the accuracy for both the Python and Julia versions of PRS-CS are nearly identical. Because the base datasets and testing populations are identical, our design allows for a head-to-head comparison of PRS-CS versus PRS.jl. Additional study limitations include the usage of a direct translation of the PRS-CS package. Although this approach allows us to directly compare the accuracy performance across the Python and Julia implementations, it does not fully use the various computational improvements that Julia affords. For example, future work aims to use Julia's multi-threading and GPU compute capabilities and are effective methods for computational acceleration of programs which heavily use matrix operations. These additional compute capabilities would allow PRS.jl to better use the hardware that users have available and make processing of even larger datasets feasible.

## Materials and Methods

### PRS.jl development

PRS-CS was cloned from https://github.com/getian107/PRScs. This implementation was translated to the Julia programming language. Development of PRS.jl was carried out in the open, with all contributions being publicly posted to the PRS.jl GitHub repository.

### Training dataset and example phenotypes

We used the Vanderbilt University Medical Center Synthetic Derivative (VUMC SD), a deidentified copy of the electronic health record (EHR), for the identification of the nine test phenotypes. VUMC is a tertiary care center that provides inpatient and outpatient care in Nashville, TN. The SD includes more than 2.8 million patient records that contain International Classification of Diseases, 9th and 10th editions (ICD-9 and ICD-10), codes; Current Procedural Terminology codes; laboratory values; medication usage; and clinical documentation (Roden et al, 2008). From the SD, a subset of patients are part of VUMC BioVU, a biobank that links the deidentified EHRs of patients to discarded blood samples for the extraction of genetic materials (Roden et al, 2008). The VUMC Institutional Review Board oversees BioVU and approved these projects.

### Genotyping and quality control

We obtained genome-wide data from 94,474 BioVU individuals genotyped on the Illumina MEGA^EX array. We used PLINK v1.9 to filter genotypes with low SNP (<0.95) call rate and individuals with low call rate (<0.98), sex discrepancies, and excessive heterozygosity (|Fhet|>0.2). Principal component analysis on the genotyped BioVU cohort together with CEU (Utah residents with Northern and Western European ancestry from the CEPH collection), YRI (Yoruba in Ibadan, Nigeria), and CHB (Han Chinese in Beijing, China) individuals from the 1000 Genomes Project Consortium et al (2015) from the CEU (Utah residents with Northern and Western European ancestry from the CEPH collection), YRI (Yoruba in Ibadan, Nigeria), and CHB (Han Chinese in Beijing, China) populations were used to create the CEU-YRI and CEU-CHB axes in FlashPCA2. Simple thresholding was used (0.3 and greater on the CEU-YRI axis and 0.4 and greater on the CEU-CHB axis) to select individuals of recent European ancestry as shown in Fig S1.

We confirmed the absence of genotyping batch effects through logistic regression with "batch" as the phenotype. We used the Michigan Imputation Server (Das et al, 2016) with the reference panel from the Haplotype Reference Consortium to impute

genotypes. SNPs were filtered for imputation quality ($R^2 > 0.3$ or INFO > 0.95) and converted to hard calls. We restricted PRS calculations to autosomal SNPs with minor allele frequency above 0.01. We removed SNPs that differed by more than 10% in minor allele frequency from the 1000 Genomes Project phase 3 CEU (1000 Genomes Project Consortium et al, 2015) set and those with a Hardy–Weinberg equilibrium $P < 10^{-10}$. The resulting data set contained hard-called SNP information for 9,386,383 SNPs in 72,828 individuals of European ancestry.

### PRS calculations

We calculate PRSs for individuals using PRS-CS (Ge et al, 2019) and our translation of the package to the Julia programming language, PRS.jl. PRS-CS/PRS.jl uses Bayesian regression with a CS before model polygenic effects on the phenotype and updates the weight of each SNP within each LD block in posterior inference. The program can use an assigned global shrinkage parameter or automatically learn the parameter from the data.

### Model performance in BioVU

Summary statistics were downloaded for six quantitative traits: body mass index, high-density lipid cholesterol, low-density lipid cholesterol, total cholesterol, triglycerides, and estimated glomerular filtration rate (Willer et al, 2013; Hellwege et al, 2019; Pulit et al, 2019) and three binary traits: asthma, coronary artery disease, and type 2 diabetes mellitus (T2DM) (Preuss et al, 2010; Zhu et al, 2019; Vujkovic et al, 2020). These traits were chosen because of their high prevalence or phenotypic validation in the VUMC EHR and usage in the original PRS-CS manuscript.

Summary statistics were processed to get these input files in a format that the original PRS-CS method can accept (columns reordered and renamed using R). PRSs were calculated in triplicate using a single-CPU architecture. Furthermore, to demonstrate the task-dependent performance improvements based on sample size, we also estimated the PRS performance for three sample sizes, 72,828 for the total population and two random subsets of the totally sized 36,000 and 18,000 individuals. Because the final sample size used in the estimate is based on the number of patients in the set with the particular outcome, we chose the most prevalent binary and continuous outcomes for this step.

The scripts used to call both programs are available at https://juliahub.com/ui/Packages/PolygenicRiskScores/zm2vm/0.1.0.

### Computational performance comparison

All computations were performed using the Vanderbilt University's Advanced Computing Center for Research and Education (ACCRE, www.accre.vanderbilt.edu). Each PRS run was restricted to a single Haswell node with an allocation of eight CPUs and 80 GB of memory. To minimize runtime variabilities related to cluster usage, we initiated the PRS-CS and PRS.jl runs for each phenotype simultaneously. Subsequent benchmarking runs were initiated over a 3-mo time course. The processing time for each PRS run was recorded. The mean and SD of the three runs per phenotype were calculated.

### PRS.jl and PRS.py performance comparison

The PRS-CS method uses a global shrinkage parameter to account for varying trait polygenicity. If a trait is highly polygenic, the global shrinkage parameter tends to be larger, whereas if a trait is less polygenic, the global shrinkage parameter will be smaller. PRS.jl and PRS-CS have two options, one of which allows the global shrinkage parameter to be automatically learned from the data rather than supplied, auto (phi). Sensitivity analyses demonstrated similar output for the two methods when using a fixed global shrinkage parameter or the auto algorithm. Thus, in each case, we used the auto version, allowing for the estimation of the global shrinkage parameter from the data. Once the posterior $\beta$ values were calculated, PLINK v1.9 was used to score each individual. PRSs for each phenotype were scaled to have mean zero and unit SD using the built-in R scale() function. Prediction accuracy was assessed using real phenotypic values in BioVU and covariate adjustment (sex, age, and PCs 1–10).

### Verification of quantitative trait performance in BioVU

Accuracy of PRSs calculated from quantitative trait summary statistics was assessed using ordinary least squares $R^2$ between the scaled PRSs, and median values by person from BioVU data processed by a previously published quality control pipeline called Quality Lab (Dennis et al, 2021).

### Verification of binary trait performance in BioVU

Because $R^2$ cannot be used for binary logistic regression, accuracy of PRSs trained from binary trait summary statistics was assessed using three measures commonly used in the PRS literature: the AUC, the Nagelkerke Pseudo $R^2$, and the odds ratio of the top 10% compared against the bottom 90% between the scaled PRSs and the binary presence of clinical codes that are representative of the clinical disease. The AUC, or area under the receiver operating characteristic curve, provides an aggregate measure that is valuable because it measures how well predictions are ranked irrespective of classification threshold. The Nagelkerke Pseudo $R^2$ is an analog of the ordinary least squares $R^2$ for logistic regression and is a commonly used to describe how well the PRS explains a binary trait (Choi et al, 2020; Maj et al, 2022). The odds ratio of the top 10% compared against the bottom 90% is a common metric used to describe how well a PRS captures the risk of developing the disease and has been used to demonstrate the validity and clinical relevance of PRSs (Khera et al, 2018). The specific codes used for asthma and coronary artery disease are available in Table S3. These codes include ICD-9 and -10 codes that mirror the clinical disease. For type 2 diabetes mellitus, however, the presence of the condition was determined using an updated version of a previously published phenotyping algorithm which is effective at distinguishing type 1 and type 2 diabetes (Pacheco & Thompson, 2012).

## Data Availability

GWAS summary statistics were downloaded from publicly available resources. BioVU summary statistics are made available upon

reasonable request to authors. PRS-CS/PRS.jl is available for download from GitHub (https://github.com/fauconab/PolygenicRiskScores.jl).

# Supplementary Information

# Acknowledgments

The work was supported by the following National Institutes of Health grants: 2R01CA157823-07A1, R01HL151152, U01HG011720, and P50MD017347. The dataset used for performance characterization was obtained from the Vanderbilt University Medical Center Synthetic Derivative, which is supported by institutional funding, the 1S10RR025141-01 instrumentation award, and by the Clinical and Translational Science Awards (CTSA) grant UL1TR000445 from National Center for Advancing Translational Sciences/ National Institutes of Health.

## Author Contributions

A Faucon: conceptualization, resources, data curation, software, formal analysis, validation, investigation, visualization, and writing—original draft, review, and editing.
J Samaroo: conceptualization, resources, data curation, software, formal analysis, validation, investigation, methodology, and writing—review and editing.
T Ge: resources, methodology, and writing—review and editing.
LK Davis: resources, data curation, supervision, funding acquisition, and writing—review and editing.
NJ Cox: conceptualization, resources, data curation, supervision, funding acquisition, and writing—original draft, review, and editing.
R Tao: resources, investigation, methodology, and writing—review and editing.
MM Shuey: conceptualization, resources, data curation, software, formal analysis, supervision, investigation, visualization, methodology, project administration, and writing—original draft, review, and editing.

## Conflict of Interest Statement

The authors declare that they have no conflict of interest.

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
