## [Reviewer comments · Life Science Alliance]

Life Science Alliance

Improving the computation efficiency of polygenic risk score modeling: Faster in Julia

Annika Faucon, Julian Samaroo, Tian Ge, Lea Davis, Nancy Cox, Ran Tao, and Megan Shuey

DOI: <https://doi.org/10.26508/lsa.202201382>

Corresponding author(s): Megan Shuey, Vanderbilt University Medical Center

Review Timeline:

Submission Date:	2022-01-21
Editorial Decision:	2022-03-08
Revision Received:	2022-06-08
Editorial Decision:	2022-06-29
Revision Received:	2022-07-05
Accepted:	2022-07-06

Scientific Editor: Novella Guidi

Transaction Report:

March 8, 2022

Re: Life Science Alliance manuscript #LSA-2022-01382-T

Megan M Shuey
Vanderbilt University Medical Center

Dear Dr. Shuey,

Thank you for submitting your manuscript entitled "Improving the computation efficiency of polygenic risk score modeling: Faster in Julia" to Life Science Alliance. The manuscript was assessed by expert reviewers, whose comments are appended to this letter. We invite you to submit a revised manuscript addressing the Reviewer comments.

Thank you for this interesting contribution to Life Science Alliance. We are looking forward to receiving your revised manuscript.

Sincerely,

B. MANUSCRIPT ORGANIZATION AND FORMATTING:

Reviewer #1 (Comments to the Authors (Required)):

Faucon and colleagues present a translation of a previously published algorithm to calculate polygenic risk into the Julia programming language, making the calculations faster. I believe this is a useful step forward, especially when the work is extended to multiple ancestries.

While I believe that this work is beneficial to the community, I am not convinced that it is a research article. The paper describes a translation of previous work into another language and provides checks to show that the results are comparable. There is no hypothesis, no application of the translated tool to a use case and no interpretation of the results. The discussion on why the speed increase is considered essential for future research is relevant, though. In short, I'm not sure if LSA is the right platform for this type of work.

I also have a couple of questions regarding the methodological details.

- Please provide the rationale for selecting the phenotypes you did.
- For the selection of the individuals that have recent European ancestry, it would be good to have a plot of the individuals and the thresholds used for inclusion.

Reviewer #2 (Comments to the Authors (Required)):

In this paper Faucon, Samaroo and colleagues present PRS.jl, a Julia translation of the Python program PRS-CS for constructing polygenic scores. By translating to a more performant language, the new method produces comparable output to PRS-CS, but runs ~5x faster. I found this paper to be simple and straightforward, and in principle it will be nice for the community to have faster methods for constructing state-of-the-art PRSs. Overall I thought that the content of the paper was sufficiently convincing, but I list a few comments below related to the presentation of the work. Overall these would result in cosmetic changes to the manuscript and so the authors should feel free to take or leave them.

-- I work in the PRS field and am quite familiar with PRS-CS, but some parts of the present manuscript seem to go into detail with little to no context. For example the second paragraph of "PRS.jl performance overview" starts with "Using the auto global shrinkage calculation with 10,000 MCMC iterations..." without explaining that the "auto" refers to performing MCMC on the global shrinkage parameter in the model, what "global shrinkage" is, or why "MCMC" is necessary. One easy fix, would be to give a 1-2 paragraph high level overview of the PRS-CS methodology and Bayesian model. On the one hand, going deep into details of PRS-CS is obviously outside the scope of this paper and not necessary, but on the other hand, readers should be able to get a sense of what is going on without needing to have both papers open side-by-side.

-- It would be nice to discuss and compare some of the implementation details in both the original Python implementation of PRS-CS and the Julia implementation. Is the performance difference entirely due to language-level differences in efficiency (i.e., is it a sort of "direct" translation) or are there changes to the algorithm that speed it up or take advantage of features of Julia? Since the major point of the paper is about the performance gain of translating PRS-CS into Julia I thought that there would be more discussion in the paper about the implementation details and the code itself.

-- Since PRS-CS is inherently stochastic one would expect different runs of PRS-CS.py to produce slightly different weights and hence PRSs. Presumably with long enough MCMC chains both should ultimately converge to the exact same posterior means and hence the same PRSs but with a finite number of MCMC iterations I imagine there's some run to run variability. I would be interested to see something like Figure 1 but comparing two runs of PRS-CS.py with different seeds, just to get a sense of whether the slight deviations seen between PRS-CS.py and PRS.jl are in line with what would be expected just from the randomness of the MCMC.

Reviewer #3 (Comments to the Authors (Required)):

The manuscript "Improving the computation efficiency of polygenic risk score modelling: Faster in Julia" by Faucon et al.

attempts to address an important issue of implementation of methods in computational biology, especially in terms of leveraging the recent improvements in terms of software techniques, in tandem with hardware developments.

Authors have reimplemented an existing method for computing the polygenic risk score (PRS-CS) and ported it from Python to Julia, a programming language particularly suitable for scientific computing. According to authors' own judgement, presented software is rather a translation than reimplementation in idiomatic Julia, and as such does not fully leverage all the opportunities allowed by the language. Nevertheless, authors present a highly useful and usable tool, which will be helpful to the community at large.

While the paper reads well and to my best knowledge is scientifically sound, there are a few questions that I believe the authors may want to consider expanding on.

Questions to the major points of the paper:

1. Task-dependence performance improvement. It is evident that performance gains are dependent on the task. It would be interesting, however, if authors could estimate how would their implementation deal with larger problem sizes, either (or both) in terms of variable count and sample count. Alternatively - if they could gauge what is the reason for a nearly two-fold dispersion in speed-ups (between 3.8x and 6.4x).
2. Interchangeability with PRS-CS. PRS.jl produces nearly identical results as the Python version, which is very good from reproducibility point of view. However, as the results are not identical, could authors comment on potential reasons for it? Would we expect similar performance in a low-data regime, that is if the number of samples (cases and controls) was lower? Especially if it was substantially lower, for example by a factor of 10 or even 50?
3. Computation efficiency profiling. Efficient computational methods, such as PRS.jl, are of great value to community. However to properly estimate the gains, it would be interesting to see what was the memory and CPU usage (in terms of core-hours) of individual runs, for both the original (Python) implementation and proposed (Julia) variant.
4. Reason for efficiency improvement. Authors report improved performance, but do not discuss the reasons for it. In Discussion (p. 10) there is an enumeration of several Julia features, which improve the computational performance of scientific code. Which of these does PRS.jl leverage? And what are the main reasons for code speedup?
5. Experimental conditions. In particular - in terms of deployment, authors paid attention to reducing the impact of heterogenous environment of computational cluster on results. However, there is still a large dispersion in terms of running times between individual experiments. What do the authors attribute it to? Have all the experiments been executed on the same hardware? If not - how comparable the hardware is and what could be the reason for difference between experiments using same data and same code?
6. Multithreading use. Authors between pages 11 and 12, write that they have yet to use multithreading capacities in Julia, while in page 14 ("Computational performance comparison") they write that execution happened on eight-core machines. Have all the cores been utilised? If so - how? If these were independent experiments running on the same machine, how did authors account for the time costs of asynchronous data access? Authors mention that cluster load may have had an impact on these metrics. It would be great if this topic could be elaborated on a bit more.

Minor issues:

page 3. "collection of risk alleles are" -> is

page 10. The first sentence of Discussion ""Polygenic risk scores are hailed..." needs reference

page 13. Acronyms "CEU-YRI" and "CEU-CHB" need to be explained.

page 17. The author of FinnGen consortium paper is currently "Consortium, F." - which is, I believe, not the intended outcome.

* Authors used R to process the data introducing a third programming language into the paper, substantially reducing the accessibility of the method. Both Python and Julia have fully fledged data processing capabilities and the part currently implemented in R could be recoded in one these languages (ideally - Julia).

* Authors use certain amount of statistical methods, which may not be intuitively familiar to the reader, including as Nagelkerke pseudo- R^2 . A sentence or two on the rationale metrics choice would be appreciated.

We would like to thank the editors and reviewers for the opportunity to submit a revised draft of our manuscript “Improving the computation efficiency of polygenic risk score modeling: Faster in Julia” to Life Science Alliance. We have incorporated the various suggestions in the revised manuscript (file name- “LSA-2022-01382-TR_final_version”) and provided responses to the reviewers’ comments below. For ease of review the changes in the marked-up manuscript (file name- “LSA-2022-01382-TR marked up copy”) are colored blue and in the responses below the corresponding text included in the manuscript are denoted as **bold**.

Editorial revisions-

We have added a summary blurb in the submission system that succinctly summarizes the study. The content of the blurb is as follows: “**To enable computationally efficient polygenic risk scores (PRSs) calculations, we translate a field standard PRS construction method, PRS-CS, to the Julia programming language.**”

We have also updated the format of the manuscript to align with the requirements for Life Science Alliance based on the manuscript preparation guidelines.

Reviewer one-

- 1) While I believe that this work is beneficial to the community, I am not convinced that it is a research article. The paper describes a translation of previous work into another language and provides checks to show that the results are comparable. There is no hypothesis, no application of the translated tool to a use case and no interpretation of the results. The discussion on why the speed increase is considered essential for future research is relevant, though. In short, I'm not sure if LSA is the right platform for this type of work.

We agree that the manuscript is not a research article and when it was originally submitted it was designated as a Research Letter prior to transfer to LSA as we acknowledge the lack of clearly testable hypotheses. We have reached out to the scientific editor of LSA, Novella Guidi, to confirm that the manuscript is within the journal’s scope. Dr. Guidi has confirmed the suitability of the journal and manuscript and agrees that it should be reassigned as a Resource Article not a Research Article. (*This designation has been updated in the online system*)

- 2) Please provide the rationale for selecting the phenotypes you did.

The rationale for phenotype choice was two-fold: 1) We wanted a set of qualitative and quantitative phenotypes to demonstrate the variable performance of the model depending on the outcome type. 2) We then prioritized phenotypes that were part of the original PRSice manuscript as we assumed readers would be familiar with the original work. The phenotypes that were most robustly characterized in the Vanderbilt University Medical Center’s electronic medical record system by participating authors were then selected for evaluation. Finally, for the one non-overlapping phenotype, estimated glomerular filtration rate, its inclusion was based on a high prevalence of the measure in our electronic medical record system and broad interest by our research entity. We have added a sentence to the Methods section under the subheading, Model Performance in

BioVU, justifying the choice. **“These traits were chosen because of their high prevalence or phenotypic validation in the VUMC EMR as well as their utilization of in the original PRS-CS manuscript.”**

- 3) For the selection of the individuals that have recent European ancestry, it would be good to have a plot of the individuals and the thresholds used for inclusion.

Thank you for this excellent suggestion. We have included the following text, **“Principal component analysis on the genotyped BioVU cohort together with individuals from 1000 genomes were used to create CEU-YRI and CEU-CHB axes in FlashPCA2 and simple thresholding was used (0.3 and greater on the CEU-YRI axis and 0.4 and greater on the CEU-CHB axis, as shown in Supplemental Figure 1) to select individuals of recent European ancestries.”** Supplemental Figure 1. We believe the provided information should sufficiently described the thresholds for determination of European genetic ancestry.

Reviewer 2-

1. I work in the PRS field and am quite familiar with PRS-CS, but some parts of the present manuscript seem to go into detail with little to no context. For example the second paragraph of "PRS.jl performance overview" starts with "Using the auto global shrinkage calculation with 10,000 MCMC iterations..." without explaining that the "auto" refers to performing MCMC on the global shrinkage parameter in the model, what "global shrinkage" is, or why "MCMC" is necessary. One easy fix, would be to give a 1-2 paragraph high level overview of the PRS-CS methodology and Bayesian model. On the one hand, going deep into details of PRS-CS is obviously outside the scope of this paper and not necessary, but on the other hand, readers should be able to get a sense of what it going on without needing to have both papers open side-by-side.

Thank you for this excellent suggestion. We have provided a high-level overview of this parameter in the methods section, subsection *PRS.jl and PRS.py Performance Comparison*. **“The polygenic risk scoring – continuous shrinkage method uses a global shrinkage parameter to account for varying trait polygenicity. If a trait is very polygenic, the global shrinkage parameter will be small, whereas if the trait is not very polygenic, the global shrinkage parameter will be large. PRS.jl and PRS-CS have two options, one of which allows the global shrinkage parameter to be automatically learned from the data rather than supplied, auto (phi).”**

2. It would be nice to discuss and compare some of the implementation details in both the original python implementation of PRS-CS and the Julia implementation. Is the performance difference entirely due to language-level differences in efficiency (i.e., is it a sort of "direct" translation) or are there changes to the algorithm that speed it up or take advantage of features of Julia? Since the major point of the paper is about the performance gain of translating PRS-CS into Julia I thought that there would be more discussion in the paper about the implementation details and the code itself.

We have provided additional text in the discussion to discuss the improvements in the Julia language that aid in the speed improvements. **“Specifically, the efficient type-system together with multiple dispatch mean that the right version of functions can be called on in a computational manner that does not require checking types at runtime. Optimized matrix routines can also prevent excess memory usage, for instance in the linkage disequilibrium table where a symmetric matrix type can be used. We also employ Julia’s multi-threading capabilities for CSV reading, along with the multi-threading built into basic linear algebra subprograms (BLAS). Employing Julia’s multithreading for additional calculations could make parts of this package, particularly everything outside of MCMC, faster.**

Further, in the discussion we had previously acknowledged that this is a basic port of the original program where-in no changes to the actual implementation script were made to improve speed, however, we did address the one change in usability we implemented beyond the direct translation which was that users can now supply summary statistics in various column orders and without column name restriction. **“Small usability improvements were made that allow the user to supply summary statistics with columns in any order and without column name restrictions. No major changes to the algorithm code were made. Thus, the improvements reflect advantages of the Julia programming language over Python.”**

3. Since PRS-CS is inherently stochastic one would expect different runs of PRS-CS.py to produce slightly different weights and hence PRSs. Presumably with long enough MCMC chains both should ultimately converge to the exact same posterior means and hence the same PRSs but with a finite number of MCMC iterations I imagine there's some run to run variability. I would be interested to see something like Figure 1 but comparing two runs of PRS-CS.py with different seeds, just to get a sense of whether the slight deviations seen between PRS-CS.py and PRS.jl are in line with what would be expected just from the randomness of the MCMC.

We have added a Supplemental table that demonstrates that the slight variations in runs using PRS-CS.py versus PRS.jl is comparable to the variations we observe across runs using the same program. These results are referenced in the Results section, **“This is similar to the median squared errors within the same program on different runs (Supplemental Table 1).”**

Reviewer 3-

1. Task-dependence performance improvement. It is evident that performance gains are dependent on the task. It would be interesting, however, if authors could estimate how would their implementation deal with larger problem sizes, either (or both) in terms of variable count and sample count. Alternatively - if they could gauge what is the reason for a nearly two-fold dispersion in speed-ups (between 3.8x and 6.4x).

Thank you for this suggestion! We added two additional sample sizes for both a binary and a continuous phenotype to discuss the performance variability based on sample count. The description of this approach is described in the Methods, **“Further to**

demonstrate the task-dependent performance improvements based on sample size, we also estimated the PRS performance for three sample sizes, 72,824 for the total population and two random subsets of the total sized 36,000 and 18,000 individuals. Because the final sample size used in the estimate is based on the number of patients in the set with the particular outcome we chose the most prevalent binary and continuous outcomes for this step.” We have also included a summarization of the results in the Results section and provided an additional Supplemental Table 2.

The variability in computational speed ups is most likely specific to the distribution of SNPs detected for each set of summary statistics. We did not investigate how such differences may affect runtime, because we expect such an analysis to be very involved and require extensive investigations into the different code paths in the library and how each set of summary statistics may affect how frequently each such code path is executed.

2. Interchangeability with PRS-CS. PRS.jl produces nearly identical results as the Python version, which is very good from reproducibility point of view. However, as the results are not identical, could authors comment on potential reasons for it? Would we expect similar performance in a low-data regime, that is if the number of samples (cases and controls) was lower? Especially if it was substantially lower, for example by a factor of 10 or even 50?

There is inherent variability in the estimates between runs . We have provided estimates for runs using fewer samples. These runs had moderately lower R^2 . The results for this reproduction based on sample size variability is provided at the end of the results section and is summarized in **Supplemental Table 2**.

3. Computation efficiency profiling. Efficient computational methods, such as PRS.jl, are of great value to community. However to properly estimate the gains, it would be interesting to see what was the memory and CPU usage (in terms of core-hours) of individual runs, for both the original (Python) implementation and proposed (Julia) variant.

Thank you for pointing this out. We did a single run while profiling average CPU utilization and maximum memory for PRS-CS and PRS.jl using a quantitative phenotype (Triglycerides) and a categorical phenotype (Asthma) using the subset of 36,000 individuals. In these benchmarks, we found some of our speed advantage explained by higher CPU utilization due to improved multi-threading. We additionally explained sources, other than greater multi-threading, for the increase in maximum allocated memory (namely describing differences in management of user-allocated memory between Julia and Python).

4. Reason for efficiency improvement. Authors report improved performance, but do not discuss the reasons for it. In Discussion (p. 10) there is an enumeration of several Julia features, which improve the computational performance of scientific code. Which of these does PRS.jl leverage? And what are the main reasons for code speedup?

We have provided additional details relating to the Julia language that allows for the computational speedups in the Discussion section, “**Specifically, the efficient type-system together with multiple dispatch mean that the right version of functions can be called on in a computational manner that does not require checking types at runtime. Optimized matrix routines can also prevent excess memory usage, for instance in the linkage disequilibrium table where a symmetric matrix type can be used. We don’t currently employ Julia’s multi-threading capabilities past what is built into basic linear algebra subprograms (BLAS) which makes both Python and Julia as fast as they are. Employing Julia’s multithreading could make parts of this package, particularly everything outside of MCMC faster.**”

5. Experimental conditions. In particular - in terms of deployment, authors paid attention to reducing the impact of heterogenous environment of computational cluster on results. However, there is still a large dispersion in terms of running times between individual experiments. What do the authors attribute it to? Have all the experiments been executed on the same hardware?

All experiments are executed using the same kind of hardware, however, the resource uses a Simple Linux Utility for Resource Management (SLURM) scheduling and managing job software package that manages jobs on large compute clusters. As such, there is some inherent variability in disk input/output (I/O) bandwidth based on data usage by other cluster users. We scheduled each set of PRS-CS and PRS.jl runs at the same time to account for some of this variability in system usage.

6. Multithreading use. Authors between pages 11 and 12, write that they have yet to use multithreading capacities in Julia, while in page 14 ("Computational performance comparison") they write that execution happened on eight-core machines. Have all the cores been utilised? If so - how?

Thank you, we have added explanation to the discussion, where we explain that we use multithreading built into Julia’s basic linear algebra subprograms (BLAS), multithreaded CSV reading, and multithreaded updates of certain MCMC variables. This allows all eight CPUs to be utilized for most portions of the MCMC calculations, but leaves room for multi-threading in other parts of the program as well.

7. Minor issues: 1) page 3. "collection of risk alleles are" -> is; 2) page 10. The first sentence of Discussion ""Polygenic risk scores are hailed..." needs reference; 3) page 13. Acronyms "CEU-YRI" and "CEU-CHB" need to be explained.; 4) page 17. The author of FinnGen consortium paper is currently "Consortium, F." - which is, I believe, not the intended outcome.

Thank you for these edits. We have made all of the suggested edits to the main document.

8. Authors used R to process the data introducing a third programming language into the paper, substantially reducing the accessibility of the method. Both Python and Julia have

fully fledged data processing capabilities and the part currently implemented in R could be recoded in one these languages (ideally - Julia).

Thank you for this excellent comment. We acknowledge that these other languages do have full fledged data processing capabilities that could be used to manage different components of the data as well as perform the statistical analyses. We chose to use R, however, as it is a common data management and statistical analysis software many researchers in the genetics field use. As such, we felt it would be most relatable to many readers as our data and statistical processing tool.

9. Authors use certain amount of statistical methods, which may not be intuitively familiar to the reader, including as Nagelkerke pseudo- R^2 . A sentence or two on the rationale metrics choice would be appreciated.

We have included additional detail in the methods section describing the various R^2 methods and indicating that these are commonly used in PRS literature and provided references. “The AUC, or area under the receiver operating characteristic curve, provides an aggregate measure that is valuable because it measures how well predictions are ranked irrespective of classification threshold (Perkins et al. 2020). The Nagelkerke Pseudo R^2 is an analog of the ordinary least squares R^2 for logistic regression, and is a commonly used to describe how well the PRS explains a binary trait (Choi et al. 2020; Perkins et al. 2020; Maj et al. 2022).”

June 29, 2022

RE: Life Science Alliance Manuscript #LSA-2022-01382-TR

Dr. Megan M Shuey
Vanderbilt University Medical Center
2525 West End Avenue, Suite 700
Nashville 37203

Dear Dr. Shuey,

Thank you for submitting your revised manuscript entitled "Improving the computation efficiency of polygenic risk score modeling: Faster in Julia". We would be happy to publish your paper in Life Science Alliance pending final revisions necessary to meet our formatting guidelines.

- please address Reviewer 2's remaining points
- please add ORCID ID for corresponding author-you should have received instructions on how to do so
- please add the Twitter handle of your host institute/organization as well as your own or/and one of the authors in our system
- please make sure that the author order in the system and in the manuscript match
- please rename the "Data Access" section to "Data Availability Statement"

A. FINAL FILES:

B. MANUSCRIPT ORGANIZATION AND FORMATTING:

Sincerely,

Reviewer #1 (Comments to the Authors (Required)):

The authors have addressed my comments sufficiently and I believe the manuscript is suitable for publication.

Reviewer #2 (Comments to the Authors (Required)):

The authors have addressed all of my previous concerns, but I have one additional comment on the newly added material. The caption and legend of Supplemental Figure 1 are insufficient to understand what is being displayed (e.g., what is "U"?). It is also a sensitive topic, particularly with the use of the word "race" in the legend. If "race" corresponds to "self-identified race", then that should be clearly stated in the figure legend and caption (although I'm not sure what it adds to the figure -- individuals could simply either be marked as included or excluded based on their position along the CEU/CHB/YRI axes). If instead "race" refers to some sort of genetic ancestry, then a more precise term should be used (see, for example, the discussion in Birney et al. <https://arxiv.org/abs/2106.10041>)

We would like to thank the editors and reviewers for their approval and additional comments for the manuscript entitled, "Improving the computation efficiency of polygenic risk score modeling: Faster in Julia" to Life Science Alliance.

We have added the following statement to the figure legend for Supplemental Figure 1 to address reviewer 2's concerns, "Individuals in the region remaining after threshold exclusion are noted by red Xs and represent the individuals included in this study. The other colors represent the administratively assigned or self-reported race for patients excluded from the study. The color key is denoted in the box in the upper right corner with the following abbreviations: B, Black or African American; W, European American or White; I, American Indian or Alaska Native; U, unknown; A, Asian; and N, Other." We have also adjusted the section title from Data Access to Data Availability Statement. Both of these changes were made in the updated .doc file for the main text.

Per the editorial guidelines we have also attached the senior authors ORCID, added a twitter statement and handles, updated the author order in the online submission, ensure the summary blurb was available, and properly formatted the main text document and figures. We hope these adjustments are in-line with the editorial requirements and will assist in rapid publication

Thank you for the opportunity to publish our work with the Life Science Alliance journal.

July 6, 2022

RE: Life Science Alliance Manuscript #LSA-2022-01382-TRR

Dr. Megan M Shuey
Vanderbilt University Medical Center
2525 West End Avenue, Suite 700
Nashville 37203

Dear Dr. Shuey,

Thank you for submitting your Research Article entitled "Improving the computation efficiency of polygenic risk score modeling: Faster in Julia". It is a pleasure to let you know that your manuscript is now accepted for publication in Life Science Alliance. Congratulations on this interesting work.

DISTRIBUTION OF MATERIALS:

Again, congratulations on a very nice paper. I hope you found the review process to be constructive and are pleased with how the manuscript was handled editorially. We look forward to future exciting submissions from your lab.

Sincerely,
